# Non-Canonical WNT/Wnt5a Pathway Activity in Circulating Monocytes of Untreated Psoriatic Patients: An Exploratory Study of Its Association with Inflammatory Cytokines and Cardiovascular Risk Marker-ADAMTS7

**DOI:** 10.3390/biomedicines11020577

**Published:** 2023-02-16

**Authors:** Claudio Karsulovic, Khanty Loyola, Raul Cabrera, Claudio Perez, Lia Hojman

**Affiliations:** 1Rheumatology Section, Internal Medicine Department, Facultad de Medicina Clínica Alemana de Santiago, Universidad del Desarrollo, Santiago 7630000, Chile; 2Investigation in Dermatology and Autoimmunity—IDeA Lab, Instituto de Ciencias e Innovación en Medicina, Universidad del Desarrollo, Santiago 7630000, Chile; 3Dermatology Section, Surgery Department, Facultad de Medicina Clínica Alemana de Santiago, Universidad del Desarrollo, Santiago 7630000, Chile; 4Cancer Immunology and Regulation Laboratory, Immunology Disciplinary Program, Facultad de Medicina, Universidad de Chile, Santiago 8380000, Chile; 5Cell Therapy Laboratory, Hospital Clínico de la Universidad de Chile, Santiago 8380000, Chile

**Keywords:** monocytes, WNT5a, cardiovascular risk, inflammation

## Abstract

The leading cause of death in psoriasis is cardiovascular disease. The determinants that induce the increase in this risk are not known. The systemic inflammatory process is dependent on lymphocytes and monocytes, as has been proposed. However, adaptation modules such as mTOR have recently been mentioned as having a role. Other factors, such as WNT and its non-canonical WNT5a-inducing pathway, are relevant in inflammation, cell migration, and neoangiogenesis. Thus, we studied circulating monocytes from untreated severe psoriatic patients and characterized inflammatory cytokines, chemokines, mTOR activity, and the cardiovascular risk marker ADAMTS7. Peripheral blood from ten severely psoriatic patients (Psoriasis severity index greater than 10) was extracted and age- and sex-matched with healthy subjects. Surface and intracellular flow cytometry were performed for cytokine, chemokine receptors, and mTOR activity. ADAMTS7 was measured using ELISA. Psoriatic patients had a higher frequency of WNT5a+ cells in monocytes, which also had higher levels of IL-1β, IL-6, CXCR3, CCR2, and phosphorylated S6R protein. We found that M1 monocytes are dominant in the WNT5a+ cell group, and intracellular levels of WNT5a were also augmented. Levels of WNT5a were correlated with ADAMTS7, a blood marker related to the pathogenesis of atheromatosis. WNT5a could be relevant to the cardiovascular risk of psoriatic patients considering its association with higher levels of inflammatory cytokines, chemokine receptors and the pro-atherogenic profile of circulating monocytes.

## 1. Introduction

Psoriasis is an autoimmune skin disease that affects up to 2–3% of the population [1]. It can affect the skin and extracutaneous territories such as the synovial, eyes, and vascular tissues [2]. The cardiovascular risk in psoriatic patients is increased, but the pathological facts that explain this risk are still under study. The systemic inflammatory process has been proposed as an essential element; however, other chronic inflammatory diseases, even those with cutaneous and musculoskeletal features, do not have this same burden. Moreover, psoriasis treatment only corrects part of the risk, which suggests the involvement of additional factors [3]. There is still a lack of a reliable methodology for measuring specific predictor factors that could be involved in the pathophysiology of the cardiovascular risk seen in psoriatic patients [4].

Monocytes and macrophages have been proposed as critical components in atherosclerotic plaque formation [5]. However, the inflammatory and adaptive processes involved are not fully identified. Th1 and Th17 inflammatory pathways and adaptation modules such as the mammalian target of rapamycin (mTOR) have been proposed as factors in accelerated plaque progression and structural blood vessel remodeling [6]. On the other hand, the cytokine microenvironment and its association with adaptation module activity such as mTOR have been intensively studied in recent years in the search for potential treatment targets [7]. Other adaptation modules, such as WNT, are also involved in tissue remodeling and local inflammatory balance. The WNT signaling pathway has a well-known role in the etiopathogenesis of cancer and some chronic inflammatory diseases, particularly in its canonical form, WNT/β-catenin [8]. When this pathway is deregulated, for example, in the instance of mutations in APC, a protein that regulates WNT activity, a higher prevalence of inherited cancers such as hereditary colorectal cancer has been described [9]. Beyond its canonical activation, there is a non-canonical pathway that drives the production of WNT5a. This glycoprotein has been associated with cell migration in chronic autoimmune processes such as psoriasis and psoriatic arthritis [10]. In this setting, the activation of WNT5a demonstrated the upregulation of proinflammatory cytokines, chemokines, and metalloproteinases, some of them associated with the transdifferentiation of myofibroblasts in the formation and maturation of atheromatous plaques [11]. In recent studies, WNT5a has been associated with other diseases with increased cardiovascular risk; its activation could be involved in the macrophage phenotyping process and changes in endothelial and muscle cells of the vascular wall [12].

Given the role of Wnt as a regulator of several processes of atheromatous plaque formation and its influence in the inflammatory features of cutaneous and non-cutaneous psoriasis, we aimed to assess the status of untreated and severely affected psoriatic patients delineating their intra-monocyte WNT5a activity and its association with specific monocyte phenotypes, chemokine-receptors expression, and a previously described circulating marker of atheromatosis, such as Disintegrin, and Metalloproteinase with Thrombospondin Motifs 7 (ADAMTS7). Establishing a three-way correlation between WNT5a activity, ADAMTS7, and monocyte inflammatory status allows us to propose early markers of cardiovascular disease from the WNT pathway in circulating monocytes.

## 2. Materials and Methods

### 2.1. Study Population

Patients were recruited among those attending the dermatology clinic at Hospital Padre Hurtado. Only naïve treatment patients were selected, with at least one year without oral treatment. A full dermatologist consultation was performed, including a Psoriasis Area Severity Index (PASI) assessment and complete clinical history. Blood samples were immediately processed using the mononuclear ficoll extraction procedure and frozen until analysis. In the second stage, we selected ten age- and gender-matched healthy controls. All participants provided informed consent following the Declaration of Helsinki. The protocol was approved by the Comité de Ética Clínica Alemana de Santiago (Acta 2020-79, approved on 15 March 2020).

### 2.2. PBMC Extraction and Flow Cytometry

Venous blood (30 mL) was obtained by cubital venopunction from all participants using BD Vacutainer 10 mL green cap heparin tubes (BD Biosciences, Franklin Lakes, NJ, USA). A PBMC Ficoll extraction was performed. Cells were washed, followed by staining with the Cell Viability Kit (BD Biosciences, Franklin Lakes, NJ, USA) and the LUNA cell counter (Brightfield—Logos Biosystems, Gyeonggi-do, Republic of Korea) with a cell viability of 98.3 ± 1.1%. PBMC were stained in duplicate with the following antibodies: Live-Dead- BUV496; V711-anti-CD3; FITC-A-anti-CD14; APC-Cy7-anti-CD16; PERCP-Cy5.5-anti-CD163; APC-A-anti-CD163; BV605-anti-human CCR2; BV786-anti-human HLA-DR; and A700-anti-CXCR3 (Biolegend, San Diego, CA, USA) at room temperature for 30 min. Finally, we fixed and permeabilized cells using a BD Cytofix/Cytoperm fixation/permeabilization kit (BD Biosciences, Franklin Lakes, NJ, USA) and stained them with Pacific Blue-anti-IL-1β, PE-Cy7-anti-human IL-6 (Biolegend, San Diego, CA, USA), A647-Phospho-S6 Ribosomal Protein (Ser235/236) Antibody (Cell Signaling Technology, Danvers, MA, USA), and PE-WNT5a Antibody (Cell Signaling Technology, Danvers, MA, USA). Monocyte subsets’ frequency and median fluorescence intensity were assessed on a FACS LSR Fortessa instrument (BD Bioscences, Franklin Lakes, NJ, USA). The data were analyzed with FlowJo software (version 10.7; TreeStar; Ashland, AZ, USA).

### 2.3. ELISA Assay

Serum ADAMTS7 levels were measured all at the same time using ELISA (Human ELISA-kit colorimetric, Novus Biologicals—Bio-Techne, Denver, CO, USA).

### 2.4. Flow Cytometry and Gating Strategy

Live cells were selected from the total number of mononuclear cells. Live cells were then negatively selected for CD3, which leaves the T lymphocyte group. The remaining cells were subsequently classified using CD14, HLA-DR, CD163, and CCR2 as surface phenotype markers (Thomas et al., 2017). Two subpopulations were defined: CD14+HLA-DR+CCR2+ cells (as M1, inflammatory cells) and CD14+CD163+CCR2- cells (as M2, non-inflammatory cells). Figure 1a. Downsampling was performed using the FlowJo plug-in available at http://www.flowjo.com (accessed on 10 August 2022).

### 2.5. Statistical Analysis

Variables are shown as means and standard deviations. Continuous variables were expressed as means, ranges, or standard deviations, and discrete variables as percentage distributions. Fisher’s exact test and Pearson’s chi-square test were used to compare categorical variables. T-test and analysis of variance test were used for continuous variables.. The Spearman’s rank correlation coefficient test analyzed the correlation between variables. All the analysis was done with GraphPad Prism version 6.01 software. Further analysis was performed using XLSTAT version 2022.2.1 (Addinsoft, New York, NY, USA). A *p*-value < 0.05 was considered statistically significant.

## 3. Results

Psoriatic patients were selected as previously described. The mean PASI was 27.3 ± 9.1, and all patients were classified as severe (above PASI 10). Even when psoriatic patients had more traditional risk factors, no significant differences were found between groups (Table 1).

First, we selected CD14+ monocytes using a negative selection gating strategy, while discarding CD3+ and CD86- cells. In these cells, intracellular levels of WNT5a in monocytes showed higher values in psoriatic patients than in the controls (Figure 1a). Then, CD14+ monocytes were gated using M1 (HLA-DR+CCR2+) and M2 (CD163+CCR-) classification. The M1/M2 ratio was then analyzed among the CD14+WNT5a+ cells. The ratio was higher in psoriatic patients (Figure 1b). Knowing that psoriatic patients had a higher frequency of WNT5a+ cells among CD14+ cells and that the M1/M2 ratio is augmented, we analyzed inflammatory cytokines, chemokine receptors, and mTOR activity. Intracellular levels of IL-1β and IL-6 were measured as being both inflammatory but through different pathways (the inflammasome and NFkB pathways, respectively). Psoriatic patients showed higher levels of both interleukins in CD14+WNT5a+ cells (Figure 1c). Similar results were found for chemokine-related proteins, CXCR3 and CCR2 (Figure 1d). For mTOR analysis, we measured intracellular levels of S6R protein phosphorylation, a crucial component of the mTOR complex 1, which participates in cell growth and protein synthesis. S6Rp values were significantly higher in WNT5a+ cells from psoriatic patients than in controls (Figure 1e).

To determine if M1 or M2 monocyte phenotypes had higher levels of WNT5a, we analyzed the median fluorescence intensity of this glycoprotein in circulating monocytes from psoriatic patients and controls. Inflammatory M1 monocytes from psoriatic patients demonstrated higher levels of WNT5a, similar to M2 monocytes (Figure 2a). Then we examined the correlation between the monocyte phenotype and the frequency of WNT5a+ cells. Only M1 correlated with WNT5a cells (Figure 2b,c). Finally, serum ADAMTS7 levels, a recently described marker of atheromatosis, correlated with WNT5a frequency, showing a positive and significant correlation (Figure 2d).

## 4. Discussion

Psoriatic patients have a higher cardiovascular risk than those with inflammatory and autoimmune diseases, even in the presence of cutaneous inflammatory features. The risk is partly associated with the clinical condition and severity of the disease; however, the magnitude of the risk cannot be predicted based on this sole aspect. Even when other inflammatory factors such as circulating cytokines, acute phase proteins, or extracutaneous involvement have been included in the analysis through new adapted scores, some guidelines have recommended correcting the individual cardiovascular risk score using mathematically based strategies using multipliers in the absence of other factors that could be involved. Although chronic inflammatory factors could be participating in an accelerated atheromatosis process, further research is needed to propose new possible elements that could be participating in the early expansion of atheromatous plaque. In this study, we first set out to establish a group with the highest risk from the skin disease point of view: those without treatment for at least one year and clinically classified as having extensive disease (measured as a PASI higher than 10). By matching them by age and sex with healthy controls, we were able to differentiate more clearly which elements could be participating in the pathophysiological process of the disease and, at the same time, be part of the pathogenesis of atheromatosis. First, in a recently published study [13], our group found that ADAMTS7 is increased in psoriatic patients and that its levels are correlated with specific cytokine and inflammatory phenotyping in circulating monocytes. In the light of these findings, we went to study the group of monocytes with positivity for the glycoprotein WNT5a, a protein of the non-canonical WNT pathway. It has recently received attention for its pro-inflammatory nature and its role in cell migration, both of which are described in the joints of autoimmune diseases such as atheromatosis. Even when the effects of WNT5a on the production of cytokines, chemokines, and metalloproteinases in macrophages have been described, there is no information on its activity and correlation with the inflammatory signature in circulating monocytes before they reach the target tissue.

Here we demonstrated that WNT5a-positive monocytes had similar behavior to that observed in macrophages, allowing us to suggest that these could bring an activation signature before their commitment to the target tissue. This will enable us to propose an answer to two new questions. Firstly, could the WNT pathway participate in at least part of the pro-atherogenic phenotype of circulating monocytes? Secondly, could the WNT pathway participate in the origin of extracutaneous involvement in psoriasis, promoting a different presentation of the disease with joint and cardiovascular involvement? It is well-known that patients with psoriatic arthritis have a higher cardiovascular risk than those without it, showing early atheromatous plaque formation and a higher incidence of cardiovascular events; nevertheless, it has been hard to demonstrate which pathophysiological elements could be interacting to start and promote this process in tissues far from the primary affected ones. Although our work involved a small number of patients and our evidence of atheromatosis is indirect, it allowed us to raise new possible factors in accelerated atheromatosis, opening up other possible areas of study in chronic inflammatory disease-associated cardiovascular risk.

## 5. Conclusions

We established a three-way correlation between inflammatory cytokines, WNT5a, and ADAMTS7 by analyzing levels of these proteins inside circulating monocytes. Beyond the correlation between them, pro-atherogenic activated proteins in circulating monocytes highlights a possible role of monocytes in the skin and non-skin manifestations and complications like early atheromatosis. 

## Figures and Tables

**Figure 1 biomedicines-11-00577-f001:**
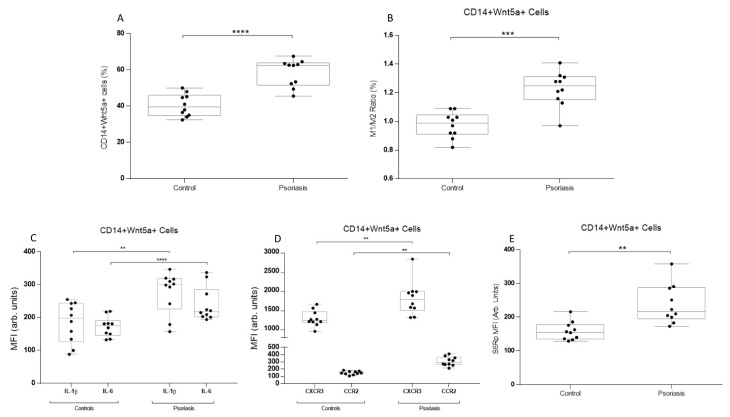
Monocyte phenotype, intracellular cytokines, and chemokine levels in CD14+Wnt5a+ cells from psoriatic patients and healthy subjects. (**A**) Frequency of CD14+Wnt5a+ cells in psoriatic patients and healthy subjects. (**B**) M1 (CD14+HLA-DR+CCR2+) monocytes phenotype and/M2 (CD14+CD163+CCR2-) monocytes phenotype ratio in CD14+Wnt5a+ cells from psoriatic patients and healthy subjects. (**C**) IL-1β and IL-6 intracellular MFI levels in CD14+Wnt5a+ cells. (**D**) CXCR3 and CCR2 intracellular MFI levels in CD14+Wnt5a+ cells. (**E**) mTOR activity marker phosphorylated S6R intracellular MFI levels in psoriatic and healthy subjects. (**A**) **** *p* = 0.0001; (**B**) *** *p* = 0.0006; (**C**) ** *p* = 0.002, **** *p* = 0.0001; (**D**) ** *p* = 0.0032; and (**E**) ** *p* = 0.0044. The Mann–Whitney test was used.

**Figure 2 biomedicines-11-00577-f002:**
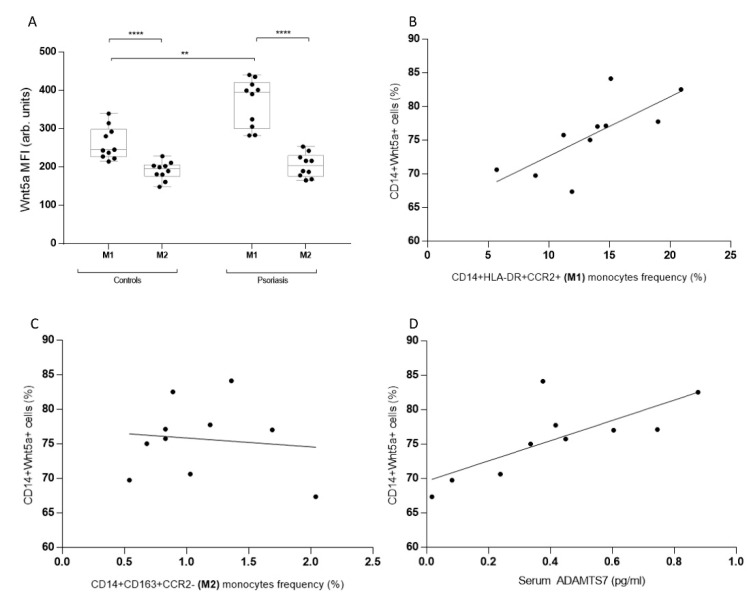
Monocyte phenotype and cardiovascular risk marker (ADAMTS7) correlations with Wnt5a levels in psoriatic and healthy subjects. (**A**) Levels of intracellular Wnt5a in M1 and M2 monocytes from healthy subjects and psoriatic patients. (**B**) CD14+Wnt5a+ cells and M1 monocytes frequency correlation in psoriatic patients. (**C**) CD14+Wnt5a+ cells and M2 monocytes frequency correlation in psoriatic patients. (**D**) CD14+Wnt5a+ cells and serum ADAMTS7 levels in psoriatic patients. (**A**) **** *p* = 0.0002, ** *p* = 0.0018; (**B**) r = 0.74, *p* = 0.015; (**C**) r = −0.11, *p* = 0.76; and (**D**) r = 0.75, *p* = 0.012. A Mann–Whitney and Spearman’s rank correlation coefficient test were both used.

**Table 1 biomedicines-11-00577-t001:** Demographic and clinical data.

	Group 1	Group 2	
	Healthy Subjects (*n* = 11)	Psoriasis (*n* = 10)	*p-*Value
Demographic			
Age, years ± SD	36.4 ± 5.7	39.1 ± 13	0.223
Men, *n*	6	6	0.81 *
Female, *n*	5	4	
Psoriasis Characteristics			
Diagnosis, years ± SD	NA	6.1 ± 9.1	
PASI, score ± SD	NA	27.3 ± 9.1	
Cardiovascular Risk Factors			
HBP, *n*	1	3	0.97 *
Lipid Disorder, *n*	1	2	
Obesity, *n*	1	2	
Diabetes, *n*	0	0	
Tobacco use, (Pack-year) ± SD	0.46 ± 1.15	3.53 ± 3.79	0.092
Medication			
Statins, *n*	1	1	

PASI: Psoriasis Area Severity Index. *: *p*-value for Pearson’s chi-square test.

## Data Availability

All data generated or analyzed during this study are included in this published article.

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
