# Peer review of "Non-Canonical WNT/Wnt5a Pathway Activity in Circulating Monocytes of Untreated Psoriatic Patients: An Exploratory Study of Its Association with Inflammatory Cytokines and Cardiovascular Risk Marker-ADAMTS7"

_biomedicines, 2023, doi:10.3390/biomedicines11020577_

Round 1

Reviewer 1 Report

1. The authors use multiple non-standard abbreviations that are not explained at first use. This does not apply to Wnt. Wnt is not an abbreviation but a portmanteau.

2. Non-insiders cannot fully understand the abstract. E.g. PASI is the Psoriasis Area Severity Index. 

3. The syntax of the last sentence of the Introduction is inaccurate and this sentence is simply incomprehensible. Help of a native English speaker in revising the manuscript is indicated. It is really hard to read the manuscript.

4. Figures are not in the right place and the graphs are not very clear because of their small size..

5. The content of this paper is rather thin or even very thin. It is a cross-sectional study with very small sample sizes. There is no clear and coherent message. There was no clear a priori hypothesis.

Reviewer 2 Report

The manuscript by Karsulovic et al. studied circulating monocytes from untreated severe psoriatic patients. Specifically, they found that higher levels of monocytes positive for non-canonical Wnt5a pathway and altered cytokine-chemokine profile  and association with atherosclerotic marker ADAMTS7. The novelty of this work is hampered by the fact that it is WNT5a was already described as an inflammatory modulator in the setting of psoriasis. Besides, the study is too preliminary and the sample size is too low to draw any conclusion.

Minor comments

-          Line 19 – ‘in’ instead of ‘is’ before ‘psoriasis’.

-          Please calculate the P-value of categorical variables of Table 1 (e.g. by Chi-square analysis).

Round 2

Reviewer 1 Report

The authors have done a significant effort to address the comments of the reviewer.

Author Response

Thank you for your time and effort in correcting our manuscript.

Reviewer 2 Report

Albeit authors have performed substantial changes in the manuscript, it is still somehow preliminary. Nevertheless, answer to the comment asking for P-value calculation of categorical variables in Table 1 has not been properly addressed. Authors have just changed NS to the P-value of age and tobacco use. Please calculate the P-value of categorical variables: men/female, HBP, lipid disorder, obesity, diabetes, and statins. It can be done by for instance Chi-square analysis or any other adequate statistical test. Thanks.

Author Response

Dear Reviewer

We apologize for this misunderstanding. Pearson's chi-square values were calculated for age-sex and cardiovascular variables an added to the table and methods as you suggested. Thank you again for your valuable time spent on our manuscript
